# Exploring Linguistic Probes for Morphological Generalization

**Jordan Kodner** and **Salam Khalifa**\* and **Sarah Payne**\*
Department of Linguistics & Institute for Advanced Computational Science
Stony Brook University, Stony Brook, NY, USA
{first.last}@stonybrook.edu

## Abstract

Modern work on the cross-linguistic computational modeling of morphological inflection has typically employed language-independent data splitting algorithms. In this paper, we supplement that approach with language-specific probes designed to test aspects of morphological generalization. Testing these probes on three morphologically distinct languages, English, Spanish, and Swahili, we find evidence that three leading morphological inflection systems employ distinct generalization strategies over conjugational classes and feature sets on both orthographic and phonologically transcribed inputs.

## 1 Introduction

The current practice in the evaluation of computational morphological inflection models, such as that employed in the SIGMORPHON, CoNLL-SIGMORPHON and SIGMORPHON-UniMorph shared tasks (Cotterell et al., 2016, 2017, 2018; McCarthy et al., 2019; Vylomova et al., 2020; Pimentel et al., 2021; Kodner et al., 2022) as well as in more targeted studies focused on specific languages or the generalization behavior of computational models (Goldman et al., 2022; Wiemerslage et al., 2022; Kodner et al., 2023b; Guriel et al., 2023; Kodner et al., 2023a), is to train on (lemma, inflection, features) triples and predict inflected forms from held-out (lemma, features) pairs. The algorithm for generating train-test splits is both random and language-independent, which has proven successful in distinguishing morphological inflection models at the gross quantitative level. Models differ in their performance across languages and in their ability to generalize across lemmas or feature sets.

In this paper, we both replicate this type of analysis and contrast it with new *language-specific probes* for testing models' generalization abilities

---
\*Denotes equal contribution

in a more controlled fashion. We examine 13 probes in three languages – English, Spanish, and Swahili – chosen for data availability and their distinct morphological characteristics. In addition, we investigate the *effect of presentation style* on performance: that is, whether the choice between phonological transcription or orthography has a substantial effect on outcomes. We report on three systems which differ widely in their behavior. They often – but not always – make reasonable linguistic generalizations, even in their incorrect predictions. In addition, we find no statistical effect for presentation style, even on English. This has implications for research that attempts to use neural networks for cognitive modeling and evaluates on orthographic data rather than more domain-appropriate phonological transcriptions.

## 2 Languages

Three languages were chosen whose inflectional morphologies range from entirely fusional (English), to mixed (Spanish), to mostly agglutinative (Swahili). In highly agglutinative languages, individual features in a set tend to correspond to distinct morphological patterns, so a model may generalize to unseen feature sets by mapping component features to their corresponding patterns. This is exemplified by the Swahili example (1), in which most features correspond to individual morphemes; only the person/number prefix maps to more than one feature. On the other hand, highly fusional languages map entire feature sets to single patterns. This is shown by the Spanish example (2), in which all features map together onto a single suffix.

(1) **Swahili *ulipika* 'you (singular) cooked'**

    *u-    li-  pik-  a*
    2.SG- PST- cook- IND

(2) **Spanish *cocinaste* 'you (singular) cooked'**

    *cocina- ste*
    cook-   2.SG.PST.IND

All data was adapted from Kodner et al. (2023b),[1] which was in turn extracted from UniMorph 3 and 4 (McCarthy et al., 2020; Batsuren et al., 2022). The data was subjected to additional processing as described below. For each language, only verbs were extracted, and multi-word expressions were excluded.[2] Importantly, no morpheme segmentation is provided in the UniMorph data unlike our illustrative examples, so agglutinativity must be discovered by each system.

**English (Germanic):** English triples were transcribed using an IPA translation of the CMU Pronouncing Dictionary.[3] Triples without available transliterations were discarded. When the dictionary provided multiple transcriptions, the lemma and inflection transcriptions which minimized Levenshtein distance between them were chosen.

**Spanish (Romance):** Spanish triples were transcribed using the Epitran package (Mortensen et al., 2018), which does not include stress information. UniMorph treats the infinitive as the lemma and includes the 2nd person plural in its paradigms.

**Swahili (Bantu):** Swahili triples were transcribed using Epitran. UniMorph treats the bare stem as the lemma. The Swahili data set is much smaller than the others and does not contain any forms with negative marking. We found many inconsistencies in the UniMorph 4 feature tags which we normalized. Most importantly, `PFV` and `PRF`, which both indicate perfect aspect, were mapped to `PFV` and tag order was made consistent. Triples in `swc.sm` with tags which could not be clearly mapped to tag sets in `swc` were excluded.

## 3 Data Splits

We created several random data splits to study dimensions of morphological generalization. As in prior work, BLIND splits were made without regard to specific linguistic properties of the triples. PROBE splits were sensitive to the properties of the triples: they were divided into relevant and irrelevant sets according to properties of the feature sets or lemmas. The irrelevant triples were split as in BLIND in order to pad training to the same length as BLIND, but irrelevant test triples were discarded. The relevant triples were split in a way specific to each probe, controlling which occurred in train+fine-tuning. For both BLIND and PROBE,

each split was performed five times with unique random seeds, and each seed was used to produce parallel orthographic and transcription versions of the splits for evaluating presentation style, yielding ten samples in total. Data sets contained 1600 training items + 400 for fine-tuning. BLIND splits contained 1000 test items.

**BLIND:** Following Kodner et al. (2023b), this splitting strategy ensures that approximately 50% of test items contain OOV feature sets but is otherwise blind to the identity of those feature sets. This was shown in the 2022 SIGMORPHON-UniMorph shared task to create more opportunities for testing dimensions of generalization across feature sets than more traditional uniform random sampling (Kodner et al., 2022).

### English PROBE Splits

**en-NFIN:** This probe is designed to test what the system does when it knows nothing about the relevant tag. The `NFIN` tag, which maps to no change, only appears in the infinitive and non-3rd singular present. Triples with the `NFIN` tag were excluded from training and presented during test. No system can know what inflection the `NFIN` corresponds to, so it can only succeed if it defaults to the lemma.

**en-PRS:** This probe tests the implications of UniMorph's design choice of annotating the non-3rd singular present with `NFIN` rather than `PRS`. We used identical splits to `en-NFIN`, but replaced the `NFIN` tag with `PRS`. Since this is shared with the present 3rd singular (`PRS;3;SG`) and participle (`V.PTCP;PRS`), a system should be reasonably expected to generalize either the *-s* or *-ing* endings to PRS items in test.

**en-PRS3SG:.** This probe also replaces `NFIN` with PRS but instead withholds `PRS;3;SG` from training and presents it during test. Success on this probe is impossible, since systems cannot learn to map this tag set to *-s* during training, but a system should generalize either the bare lemma or the *-ing* of the present participle.

### Spanish PROBE Splits

**es-FUT:** This probe tests the ability of systems to learn a basic agglutinative suffixing pattern: the future tense (`IND;FUT`) is formed by suffixing regular person/number marking onto the infinitive (*correr* 'to run,' *correr-ás* 'you will run'). There are six possible person/number combinations attested in the future indicative paradigm. For each split/seed, two feature sets with `IND;FUT` were randomly cho-

---
[1]https://github.com/jkodner05/ACL2023_RealityCheck.
[2]https://github.com/jkodner05/EMNLP2023_LingProbes with summaries provided in the Appendix.
[3]https://github.com/menelik3/cmudict-ipa.

sen, and then triples with these features were randomly sampled to appear in training for models to learn the pattern. All other feature sets containing `IND;FUT` were withheld from training, and triples containing them were sampled to appear in test.[4]

**es-AGGL:** The conditional (`COND`) and imperfect (`IND;PST;IPFV`) are also agglutinative. Two each of conditional, imperfect, and future feature sets were independently selected to appear in the training split, and the rest were withheld for evaluation. This is a more challenging version of the `es-FUT` probe because a system has to learn three agglutinative patterns at once.

**es-PSTPFV:** This probe tests what a system does when it is forced to predict missing parts of a fusional paradigm. As shown in (2), preterite (`IND;PST;PFV`) person/number forms are fusional, because they manifest distinctly from the marking for all other tense/aspects (except for the 1st person plural) and are not decomposable into a preterite part and a person/number part. Since a system cannot predict these forms from their component person/number features matched with other tenses, it should perform poorly even if it succeeds at `es-FUT` and `es-AGGL`. A system should generalize person/number marking from the other tenses.

**es-IR:** This probe tests generalization across conjugational classes rather than across feature sets. Spanish verbs fall into three classes clearly indicated by their infinitive suffix: *-ar*, *-er*, and *-ir*. *-ir* shares many, but not all, of its inflections with *-er*. For this probe, all but 50 randomly chosen *-ir* lemmas are banned from train sampling, which results in 10-18 *-ir* training triples sampled per seed. A system should predict the appropriate *-er* form or one with *-i-* for each *-ir* evaluation item or overapply the majority *-ar* ending.

**es-IRAR:** This probe is similar to `es-IR` but much more challenging because *all -ar* and *-ir* verbs are withheld from training. We predict that a system should produce the *-er* inflected form or replace *-e-* in forms with the appropriate *-a-* or *-i-*.

**Swahili PROBE Splits**

**sw-1PL:** As in (1), person/number and tense/aspect marking is marked agglutinatively before the stem in Swahili. The 1st person plural (`1;PL`) is marked with a *tu-* prefix. Two feature sets containing `1;PL` were randomly selected by seed and allowed to appear in train, and the rest were withheld for eval-

uation. This test is similar to `es-FUT` and `es-AGGL`.

**sw-NON3:** Swahili manifests four non-3rd person subject marking prefixes as well as a 3rd person prefix for each of its many noun classes. This is a more challenging version of `sw-1PL` which withholds all but one independently selected feature set containing each of `1;SG`, `1;PL`, `2;SG`, and `2;PL` from training and evaluates on the rest.

**sw-FUT:** Tense is marked with an affix immediately following subject marking. The future is marked with *-ta-*. This probe is set up like `sw-1PL` except it requires a system to produce a string infix rather than string prefix.

**sw-PST:** The simple past (`PST`) is marked with *-li-*. This probe is similar to `sw-FUT` but with a distractor: the past perfective (`PST;PFV`) is actually fusional, and is expressed as *-me-* without *-li-*. A system could thus produce *-me-* forms instead of the expected *-li-*.

**sw-PSTPFV:** This probe is similar to `sw-PST` except it tests the past perfective (`PST;PFV`), while the simple past (`PST`) serves a distractor.

## 4 Systems

**CHR-TRM** (Wu et al., 2021) is a character-level transformer that was used as a baseline in the 2021 and 2022 SIGMORPHON shared tasks. We use default hyper-parameters for low-resource settings.

**CLUZH** (Wehrli et al., 2022) is a character-level transducer which performs well but showed some weakness in feature set generalization in the 2022 shared task. We used beam decoding, size = 4.

**ENC-DEC** (Kirov and Cotterell, 2018) is an LSTM-based encoder-decoder which was argued to provide evidence for the cognitive plausibility of connectionist models as a follow-up to the Past Tense Debate.

## 5 Results

### 5.1 Orthography and Transcription

This section analyzes the effect of presentation style on performance. In addition to visual inspection of Figure 1, which shows little difference between orthography and phonological transcription, there are at most moderate differences in mean accuracy between the two. Differences range from +4.07 points in favor of orthography for English, to -2.80 for Spanish, to only -0.45 for Swahili.

English may favor orthography because it removes the three-way allomorphy of past *-(e)d* and

---

[4]Illustrations for two seeds are provided in the Appendix.

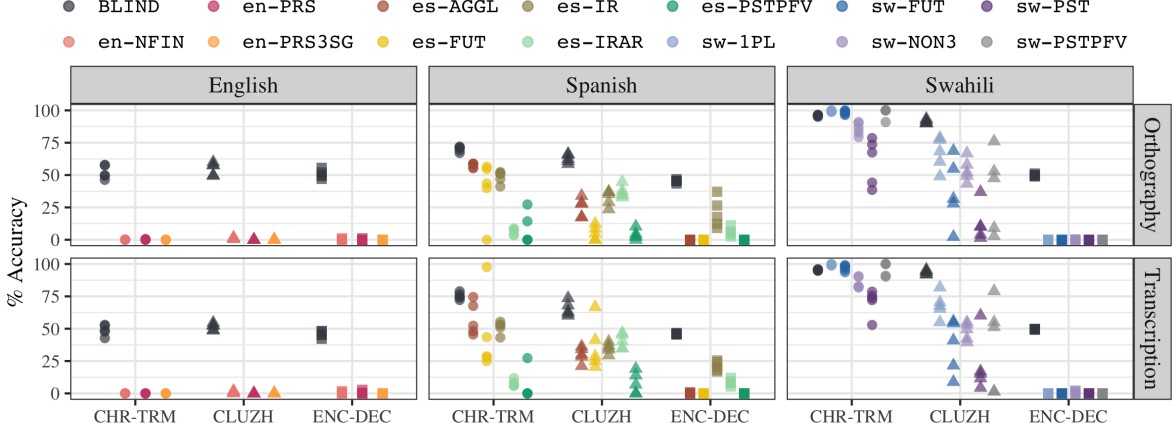

Figure 1: Accuracy on each split (color) for each seed by language (major column), system (minor column/shape), and presentation style (row).

3rd singular present *-(e)s* which is indicated in transcription. However, it is also possible that the difference is due to the choice of transcription dictionary or how we processed it. For Spanish, a transcription scheme that retained stress may have proven more challenging than Epitran's which lacks it.

| Variable | F-statistic | *p*-value |
|---|---|---|
| **system** | **68.093** | **<2e-16** |
| seed | 0.223 | 0.925 |
| presentation | 0.014 | 0.906 |
| **language** | **76.588** | **<2e-16** |
| presentation * lang | 1.061 | 0.351 |

Table 1: ANOVA analysis on BLIND showing significant effect for system and language but not presentation.

We follow this with an ANOVA analysis of five variables: the system, seed, presentation style, language, and the interaction between presentation and language, to determine which differences in mean accuracy are unlikely to be due to chance. Summarized in Table 1, we find that only the choice of system and the language are significant, not presentation style. This is consistent with visual inspection. The conclusion is the same on BLIND, PROBE, and both combined.

## 5.2 Generalization and Linguistic Analysis

**Feature Set Generalization in BLIND**

A breakdown of BLIND test triple by type in Table 2 replicates prior work (Kodner et al., 2022, 2023b) demonstrating that generalization to unseen feature sets is particularly challenging. All systems showed lower accuracy on OOV feature sets (fsOOV & bothOOV) than on other triples. ENC-DEC shows virtually no ability to do this.[5]

<hr/>

[5] A complete breakdown is provided in the Appendix.

| System | noOOV | lmOOV | fsOOV | bothOOV |
|---|---|---|---|---|
| CHR-TRM | 94.40% | 82.68% | 52.90% | 36.46% |
| CLUZH | 93.93 | 95.43 | 47.12 | 48.93 |
| ENC-DEC | 93.79 | 86.01 | 2.53 | 1.43 |

Table 2: Average performance on BLIND orthography across seeds and languages. noOOV = triples where both lemma and feat. set were observed; lmOOV = lemma is OOV; fsOOV = feature set is OOV. bothOOV = lemma and feature set are OOV.

For CHR-TRM and CLUZH, we observe a much smaller performance gap on OOV feature sets for the more agglutinative languages (Swahili -9.5 points < Spanish -49.76 < English -81.38), which indicates that these systems can perform some degree of generalization across feature sets. This contradicts the prior work, which uncovered no substantial difference between fusional and agglutinative languages, indicating an inability to perform this kind of generalization.

This discrepancy may be explained by how we processed the Swahili data. The published UniMorph data contains several inconsistencies, summarized in Section 2, in its feature tags which made the task of generalization unfairly challenging in prior work. We corrected this by normalizing the feature sets and removing triples which we could not adequately fix. However, this would only explain the middling reported performance on other agglutinative languages if their UniMorph data sets turn out to be highly inconsistent as well. This question is left for future work.

**Generalization on Swahili PROBE Splits**

CHR-TRM is very successful at generalizing the prefix in sw-1PL and string infix in sw-FUT. CLUZH sometimes applies the 1st singular prefix instead

of the plural in `sw-1PL`, but is less consistent for `sw-FUT`, where it produces many infixes belonging to other tenses in addition to nonsense outputs. `sw-NON3` was developed as a more challenging version of `sw-1PL`, and performance was indeed lower. However, instead of "near-miss" errors suggesting linguistic generalization, both systems produced a range of person/number, tense, and nonsense errors not clearly related to the probe.

`sw-PST` was designed to be similar to `sw-FUT` but with a distractor fusional string infix. As expected, performance was lower, especially for CLUZH, which often substituted the distractor past perfect, other tense marking, or omitted tense altogether similar to its errors on `sw-FUT`. Its errors in `sw-PSTPFV` were overwhelmingly application of the simple past, which can be explained as a generalization of PST while ignoring the PFV tag. These are mostly reasonable errors that point to some degree of generalization. ENC-DEC showed no ability to generalize according to component morphological features even this highly agglutinative setting.

## Generalization on Spanish PROBE Splits

For `es-FUT`, CHR-TRM and CLUZH identified the correct pattern of infinitive + person/number marking but often produced slightly incorrect forms. In the orthographic tests, many of these errors involved incorrect stress marking. These could be considered reasonable "near-misses" that point to generalization. However, for the more challenging `es-AGGL`, both produced less interpretable errors. `es-PSTPFV` was designed to be impossible but insightful, since preterite person/number marking cannot be predicted from the person/number marking of other tenses. Indeed, CLUZH generalized the person/number marking from the other tenses as well as the few preterite person/number markings presented during training, indicating that it can employ the relevant generalization. However, CHR-TRM and ENC-DEC were less interpretable.

`es-IR` and `es-IRAR` differ from the other probes in that they require generalization across conjugational classes indicated by the form of the lemma rather than generalization across feature sets. This proved appropriately challenging for all systems, which all produced many nonsense errors and some near-miss overuse of *-er* endings or combined *-a+er* and *-i+er* endings. Notably, these were also the only probes for which ENC-DEC showed some success. The tasks that the system was asked to

perform in (Kirov and Cotterell, 2018) rewarded analogy across lemma forms rather than unseen feature sets. This is similar to the `es-IR` and `es-IRAR` probes but distinct from all the other probes which were focused on feature generalization.

## Generalization on English PROBE Splits

Accuracy was near-zero on every English PROBE. This was expected, because the tasks were effectively impossible, however, qualitative analysis is insightful. For `en-NFIN` and `en-PRS`, a system would succeed if it defaulted to the bare lemma as an inflection of last resort. However, no system took that approach. For `en-NFIN`, CHR-TRM consistently outputted *-ing* forms with other errors, CLUZH produced *-(e)d* forms, and ENC-DEC produced mostly *-(e)s* forms with other errors.

The replacement of UniMorph's NFIN with PRS did indeed have an effect, indicating that some behavior on English can be attributed to this corpus design choice: Both CHR-TRM and CLUZH now consistently produced present participle *-ing* forms, while ENC-DEC instead produced *-(e)d* or *-(e)s*. This indicates generalization over the PRS feature. For `en-PRS3SG`, CLUZH always produced the bare lemma form, showing clear generalization of PRS. CHR-TRM only produced nonsense errors, while ENC-DEC produced nonsense errors suffixed with *-(e)d* or *-ing*.

## 6   Conclusions

In this work, we present language-specific probes to evaluate the ability of computational systems to perform morphological generalizations as a complement to prior work relying on large-scale language-independent data-splitting. Systems differ substantially both in their ability to perform generalizations and in their problem-solving strategies. Of the two systems showing generalization ability, both perform well on the simplest probes but struggle on more complex but feasible probes. On probes designed to be impossible but insightful, both systems show some degree of reasonable generalization. In addition to this, we find no significant effect of presentation style on the behavior of the three systems. While we maintain that the choice of presentation style should be driven by application when possible (usually orthography for NLP or transcription for cognitive modeling), our results suggest that it can be reasonable to use orthography when transcription is impractical.

## Limitations

The main limitation of the the language-specific splitting approach compared to the traditional language-independent splitting approach is the language-specific and domain-specific expertise needed to design the probes themselves. Nevertheless, this work successfully demonstrates the utility of such probes. Future work with a larger group of experts in a wider range of languages could extend this approach to more test cases. We believe that this could be implemented on a larger scale such as for a complementary shared task in the future.

## Ethics Statement

To the best of our knowledge, all results published in this paper are accurate. All data sources are free, publicly available, and cited in the article. No sensitive data was used which could violate individuals' privacy or confidentiality. Authorship and acknowledgements fairly reflect contributions.

## Acknowledgements

We thank Zoey Liu for motivating discussion. Experiments were performed on the SeaWulf HPC cluster maintained by RCC and the Institute for Advanced Computational Science (IACS) at Stony Brook University and made possible by National Science Foundation (NSF) grant No. 1531492. The third author gratefully acknowledges funding through the IACS Graduate Research Fellowship and the NSF Graduate Research Fellowship Program under NSF Grant No. 2234683.

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

# A Appendix

|          | # Lemmas | # Feature Sets | # Triples |
|----------|----------|----------------|-----------|
| English  | 9188     | 5              | 27836     |
| Spanish  | 7326     | 152            | 1077655   |
| Swahili  | 131      | 169            | 10925     |

Table 3: Type frequencies for lemmas, feature sets, and triples for each language data set before splitting.

| Ortho | noOOV | lmOOV | fsOOV | bothOOV | Total |
|-------|-------|-------|-------|---------|-------|
| CHR-TRM | 95.83% | 14.32% | 95.32% | 7.67% | 52.22% |
|  | *(11.76)* | *(39.60)* | *(7.34)* | *(20.30)* | *(11.60)* |
| CLUZH | 96.60% | 20.43% | 95.28% | 12.85% | 54.96% |
|  | *(7.84)* | *(39.60)* | *(7.83)* | *(24.00)* | *(10.60)* |
| ENC-DEC | 97.22% | 8.58% | 95.37% | 5.24% | 50.76% |
|  | *(7.84)* | *(31.68)* | *(7.59)* | *(15.79)* | *(8.50)* |
| **Transcr** | **noOOV** | **lmOOV** | **fsOOV** | **bothOOV** | **Total** |
| CHR-TRM | 83.44% | 14.08% | 89.86% | 7.04% | 48.86% |
|  | *(25.72)* | *(38.38)* | *(11.67)* | *(17.23)* | *(10.30)* |
| CLUZH | 85.77% | 18.10% | 90.59% | 11.44% | 51.48% |
|  | *(13.77)* | *(38.38)* | *(14.38)* | *(20.12)* | *(6.00)* |
| ENC-DEC | 85.44% | 4.12% | 89.22% | 2.37% | 45.82% |
|  | *(26.03)* | *(12.12)* | *(13.64)* | *(6.08)* | *(6.00)* |

Table 4: Average percent accuracy and (*accuracy range*) on BLIND English across seeds for each system and presentation style. noOOV = triples where both lemma and feature set were observed; lmOOV = lemma is OOV; fsOOV = feature set is OOV. bothOOV = lemma and feature set are OOV.

| Ortho | noOOV | lmOOV | fsOOV | bothOOV | Total |
|-------|-------|-------|-------|---------|-------|
| CHR-TRM | 93.38% | 48.14% | 91.92% | 47.98% | 70.16% |
|  | *(6.87)* | *(18.41)* | *(1.67)* | *(5.43)* | *(4.90)* |
| CLUZH | 91.96% | 33.14% | 93.38% | 32.05% | 62.74% |
|  | *(9.63)* | *(21.45)* | *(2.45)* | *(13.06)* | *(7.80)* |
| ENC-DEC | 90.54% | 0.59% | 90.27% | 0.25% | 45.34% |
|  | *(9.56)* | *(2.97)* | *(5.55)* | *(0.51)* | *(3.30)* |
| **Transcr** | **noOOV** | **lmOOV** | **fsOOV** | **bothOOV** | **Total** |
| CHR-TRM | 95.55% | 56.40% | 93.97% | 56.08% | 75.26% |
|  | *(7.27)* | *(29.78)* | *(3.03)* | *(7.44)* | *(6.70)* |
| CLUZH | 92.22% | 38.37% | 93.37% | 37.21% | 65.32% |
|  | *(10.41)* | *(29.89)* | *(3.35)* | *(25.79)* | *(13.50)* |
| ENC-DEC | 92.01% | 0.39% | 91.20% | 0.71% | 46.04% |
|  | *(7.46)* | *(1.96)* | *(3.54)* | *(2.78)* | *(1.20)* |

Table 5: Average percent accuracy and (*accuracy range*) on BLIND Spanish across seeds for each system and presentation style. noOOV = triples where both lemma and feature set were observed; lmOOV = lemma is OOV; fsOOV = feature set is OOV. bothOOV = lemma and feature set are OOV.

| Ortho | noOOV | lmOOV | fsOOV | bothOOV | Total |
|-------|-------|-------|-------|---------|-------|
| CHR-TRM | 99.16% | 92.75% | 75% | 50.00% | 95.92% |
|  | *(1.40)* | *(3.40)* | *(50.00)* | *(100)* | *(1.60)* |
| CLUZH | 98.48% | 84.14% | 100% | 100% | 91.32% |
|  | *(0.99)* | *(6.35)* | *(0.00)* | *(0.00)* | *(3.50)* |
| ENC-DEC | 98.80% | 1.00% | 75.00% | 0.00% | 49.88% |
|  | *(2.20)* | *(3.20)* | *(50.00)* | *(0.00)* | *(2.30)* |
| **Transcr** | **noOOV** | **lmOOV** | **fsOOV** | **bothOOV** | **Total** |
| CHR-TRM | 99.04% | 91.71% | 50.00% | 50.00% | 95.32% |
|  | *(1.60)* | *(3.22)* | *(100)* | *(100)* | *(1.10)* |
| CLUZH | 98.56% | 88.55% | 100% | 100% | 93.56% |
|  | *(1.20)* | *(6.78)* | *(0.00)* | *(1.20)* | *(3.70)* |
| ENC-DEC | 98.76% | 0.48% | 75.00% | 0.00% | 49.60% |
|  | *(1.80)* | *(1.60)* | *(50.00)* | *(0.00)* | *(0.60)* |

Table 6: Average percent accuracy and (*accuracy range*) on BLIND Swahili across seeds for each system and presentation style. noOOV = triples where both lemma and feature set were observed; lmOOV = lemma is OOV; fsOOV = feature set is OOV. bothOOV = lemma and feature set are OOV.

| Ortho | en-NFIN | en-PRS | en-PRS3SG |
|-------|---------|--------|-----------|
| CHR-TRM | 0.06% | 0.18% | 0.00% |
|  | *(0.29)* | *(0.60)* | *(0.00)* |
| CLUZH | 1.11% | 0% | 0.09% |
|  | *(1.24)* | *(0.00)* | *(0.47)* |
| ENC-DEC | 0.35% | 0.23% | 0.00% |
|  | *(1.17)* | *(1.15)* | *(0.00)* |
| **Transcr** | **en-NFIN** | **en-PRS** | **en-PRS3SG** |
| CHR-TRM | 0.00% | 0.00% | 0.00% |
|  | *(0.00)* | *(0.00)* | *(0.00)* |
| CLUZH | 0.92% | 0.00% | 0.09% |
|  | *(2.05)* | *(0.00)* | *(0.47)* |
| ENC-DEC | 0.63% | 0.67% | 0.00% |
|  | *(1.69)* | *(2.79)* | *(0.00)* |

Table 7: Percent accuracy and (*accuracy range*) on English PROBE splits by system and presentation style.

| Ortho | es-AGGL | es-FUT | es-PSTPFV | es-IR | es-IRAR |
|-------|---------|--------|-----------|-------|---------|
| CHR-TRM | 57.47% | 38.93% | 8.31% | 48.61% | 6.32% |
|  | *(3.54)* | *(56.41)* | *(27.27)* | *(11.13)* | *(5.42)* |
| CLUZH | 24.99% | 5.19% | 3.94% | 32.39% | 37.05% |
|  | *(16.64)* | *(12.50)* | *(10.26)* | *(13.57)* | *(12.09)* |
| ENC-DEC | 0.00% | 0.00% | 0.00% | 20.52% | 5.95% |
|  | *(0.00)* | *(0.00)* | *(0.00)* | *(28.01)* | *(8.76)* |
| **Transcr** | **es-AGGL** | **es-FUT** | **es-PSTPFV** | **es-IR** | **es-IRAR** |
| CHR-TRM | 57.61% | 44.57% | 5.45% | 51.12% | 7.86% |
|  | *(28.98)* | *(72.62)* | *(27.27)* | *(12.25)* | *(6.05)* |
| CLUZH | 30.05% | 36.41% | 7.95% | 35.13% | 40.36% |
|  | *(15.12)* | *(46.15)* | *(19.05)* | *(10.59)* | *(12.08)* |
| ENC-DEC | 0.19% | 0.00% | 0.00% | 21.08% | 8.55% |
|  | *(0.96)* | *(0.00)* | *(0.00)* | *(8.57)* | *(6.69)* |

Table 8: Percent accuracy and (*accuracy range*) on Spanish PROBE splits by system and presentation style.

| Ortho | sw-1PL | sw-NON3 | sw-FUT | sw-PST | sw-PSTPFV |
|-------|--------|---------|--------|--------|-----------|
| CHR-TRM | 99.37% | 85.90% | 98.35% | 60.48% | 98.19% |
|  | *(1.14)* | *(11.71)* | *(3.60)* | *(40.12)* | *(9.03)* |
| CLUZH | 66.71% | 53.89% | 37.10% | 12.56% | 37.66% |
|  | *(29.70)* | *(23.52)* | *(66.49)* | *(35.44)* | *(73.24)* |
| ENC-DEC | 0.00% | 0.05% | 0.00% | 0.00% | 0.00% |
|  | *(0.00)* | *(0.24)* | *(0.00)* | *(0.00)* | *(0.00)* |
| **Transcr** | **sw-1PL** | **sw-NON3** | **sw-FUT** | **sw-PST** | **sw-PSTPFV** |
| CHR-TRM | 99.39% | 85.56% | 96.94% | 70.56% | 96.24% |
|  | *(1.14)* | *(8.70)* | *(5.38)* | *(25.74)* | *(9.79)* |
| CLUZH | 68.00% | 47.94% | 36.24% | 21.68% | 37.61% |
|  | *(26.82)* | *(15.57)* | *(46.83)* | *(55.85)* | *(77.64)* |
| ENC-DEC | 0.00% | 0.81% | 0.00% | 0.00% | 0.00% |
|  | *(0.00)* | *(2.16)* | *(0.00)* | *(0.00)* | *(0.00)* |

Table 9: Percent accuracy and (*accuracy range*) on Swahili PROBE splits by system and presentation style.

| es-FUT Seed 0 | |
|---|---|
| Train/FTune-Only Feature Sets | `V;IND;FUT;2;SG;FORM` |
| | `V;IND;FUT;3;SG` |
| Test-Only Feature Sets | `V;IND;FUT;1;PL` |
| | `V;IND;FUT;1;SG` |
| | `V;IND;FUT;2;SG;INFM` |
| | `V;IND;FUT;2;PL;INFM` |
| | `V;IND;FUT;3;PL` |
| # Train-Only Triples Sampled | 38 |
| # Other Train/FTune Triples Sampled | 1962 |
| # Test-Only Triples Sampled | 46 |
| # Unique Train/FT-Only F. Sets Sampled | 2 |
| # Unique Other Train/FT F. Sets Sampled | 145 |
| # Unique Test-Only F. Sets Sampled | 5 |

| es-FUT Seed 4 | |
|---|---|
| Train/FTune-Only Feature Sets | `V;IND;FUT;1;SG` |
| | `V;IND;FUT;2;PL;INFM` |
| Test-Only Feature Sets | `V;IND;FUT;1;PL` |
| | `V;IND;FUT;2;SG;INFM` |
| | `V;IND;FUT;2;SG;FORM` |
| | `V;IND;FUT;3;SG` |
| | `V;IND;FUT;3;PL` |
| # Train/FTune-Only Triples Sampled | 34 |
| # Other Train/FTune Triples Sampled | 1966 |
| # Test-Only Triples Sampled | 42 |
| # Unique Train/FT-Only F. Sets Sampled | 2 |
| # Unique Other Train/FT F. Sets Sampled | 145 |
| # Unique Test-Only F. Sets Sampled | 5 |

Table 10: Description of `es-FUT` train and test feature sets for seeds 0 and 4. Seeds 1-3 provide similar results, and similar splitting logic applies to the other PROBE splits. Triples with feature sets containing `FUT` are partitioned at random by seed into those that can only be sampled for train+finetune and those that can only be sampled for test. Otherwise, data is split uniformly at random as in most SIGMORPHON shared tasks. No relevant PROBE feature set appears in both train and test, but all other feature sets may. Evaluation is only performed on sampled triples with the relevant PROBE feature set. This approach allows us to test the specific impact of the PROBE in an otherwise typical setting.