# OpenReview forum: "Exploring Linguistic Probes for Morphological Inflection"
_EMNLP/2023/Conference — EMNLP 2023 Main_

### Official Review · Reviewer_FThw · 2023-07-24

**Soundness:** 5

**Excitement:**

5: Transformative: This paper is likely to change its subfield or computational linguistics broadly. It should be considered for a best paper award. This paper changes the current understanding of some phenomenon, shows a widely held practice to be erroneous in someway, enables a promising direction of research for a (broad or narrow) topic, or creates an exciting new technique.

**Paper Topic And Main Contributions:**

This short paper designs and presents linguistically-motivated morphological probes for inflectional morphology models. The authors observe that most prior work relies on language-independent probes while they would like to see how well the models do with respect to common language specific features, e.g. the fact that in Spanish, -ir verbs share some but not all inflections with -er verbs; the ways Spanish tenses are formed, and so forth. They design probes for three languages: English (little morphology; fusional), Spanish (rich morphology; fusional), and Swahili (rich morphology; agglutinative). They hypothesize that some probes will be more challenging than the others, and show that to be the case.

**Reasons To Accept:**

The paper is very interesting to me because it is linguistically informed. It provides a detailed and meaningful description of the data and methods. The topic falls into the category that many consider the most important today: how well models encode language structure. It goes beyond most papers that are published on this topic in that the design is actually linguistically meaningful. The authors provide insightful and detailed comments on their results.

**Reasons To Reject:**

I would like to see this paper accepted. The only thing I object to it is the somewhat unnecessary use of the term "cognitive" a couple times; so far nobody has really shown that any of the LMs encode anything truly "cognitive", and for the given paper, it doesn't seem necessary to repeat that dubious term.

**Reproducibility:**

5: Could easily reproduce the results.

**Reviewer Confidence:**

4: Quite sure. I tried to check the important points carefully. It's unlikely, though conceivable, that I missed something that should affect my ratings.

**Typos Grammar Style And Presentation Improvements:**

It took me a bit to figure out how to read the results Figure (Figure 1). Maybe a comment on the different icon shapes would help?

---

> ### Author Rebuttal · Authors · 2023-08-26
>
> We fully agree with the reviewer here that the term "cognitive" should not be thrown around lightly, and that it hasn't really been demonstrated that these LMs make appropriate cognitive models. Our use of the term was meant to convey the position taken in some of the literature that attempts to use NNs for cognitive modeling. For example, the paper that introduces our ENC-DEC model explicitly positions itself as a cognitive modeling paper. We will either clarify this in the text or just remove the term altogether as the reviewer suggests.
>
> We will find a way to make Figure 1 clearer. The shapes are meant to be visually redundant ways to differentiate the models, but if that's confusing, we don't necessarily need them, since the models are already grouped into sub-columns. In any case, we plan on expanding the Fig 1 caption with a lot more explanation on how to read the plot if the paper is accepted.

---

### Official Review · Reviewer_gRQQ · 2023-08-02

**Soundness:** 4

**Excitement:**

4: Strong: This paper deepens the understanding of some phenomenon or lowers the barriers to an existing research direction.

**Paper Topic And Main Contributions:**

This paper introduces language-specific linguistic probes to assess the generalization of morphological inflection systems in English, Spanish, and Swahili. The probes are designed to be meaningful, considering linguistic information, providing a novel approach compared to previous language-independent methods. The study explores the impact of ambiguous examples and evaluates model performance on out-of-vocabulary (OOV) data. Additionally, the paper compares the use of orthography and phonological transcriptions, concluding that there are not significand differences. Statistical analysis confirms the significance of model and language factors on the results. Results prove the usefulness of the proposed probes to test the strengths and weaknesses of morphological inflection models.

**Questions For The Authors:**

1. Why did you select these specific probes? Explanations are very condensed (I understand the lack of space in a short paper) and they are hard to understand and to know the motives behind them.

**Reasons To Accept:**

- Language-specific, linguistic-motivated experiments.
- Include 3 languages with different typology.
- Statistical and error analysis.
- Clear and concise writing.

**Reasons To Reject:**

None

**Reproducibility:**

4: Could mostly reproduce the results, but there may be some variation because of sample variance or minor variations in their interpretation of the protocol or method.

**Reviewer Confidence:**

3: Pretty sure, but there's a chance I missed something. Although I have a good feel for this area in general, I did not carefully check the paper's details, e.g., the math, experimental design, or novelty.

**Typos Grammar Style And Presentation Improvements:**

Explanation of probes is a bit hard to follow. It would be helpful to have some qualitative examples on a Figure to make it easy to have an idea of what processes are being test by the probes.

---

> ### Author Rebuttal · Authors · 2023-08-26
>
> Thank you for your review.
>
> We agree that the explanations of the probes can be improved and we should take the space to explain why we chose them (It's tricky getting everything into 4 pages for a short paper submission). If the paper is accepted, we intend to use most of the 5th page for expanding the explanations of the probes and expanding on the analysis. That should improve clarity substantially.

---

### Official Review · Reviewer_1U8W · 2023-08-04

**Soundness:** 4

**Excitement:**

5: Transformative: This paper is likely to change its subfield or computational linguistics broadly. It should be considered for a best paper award. This paper changes the current understanding of some phenomenon, shows a widely held practice to be erroneous in someway, enables a promising direction of research for a (broad or narrow) topic, or creates an exciting new technique.

**Paper Topic And Main Contributions:**

The paper proposes a set of 13 language-specific probes for English, Spanish, and Swahili and evaluate the ability of 3 different models from prior work to generalize morphological patterns. This work supplements prior work on morphological inflection, which focuses on language-independent data splitting.

**Questions For The Authors:**

* line 240: which feature tags inconsistencies in the UniMorph data are you referring to?
* Limitations Section: you mention that future work includes a 'larger group of experts'. How many experts were involved in the current experiments? or any relevant information
* Appendix A (Tables 8,9): The model ENC-DEC performs significantly worse than the other two, achieving 0% accuracy on most of the Spanish and Swahili probes. Are there any insights as to why it underperforms in languages other than English? Also, does this suggest that character-level models are generally better at morphological generalization?

**Reasons To Accept:**

* The paper is well written. The method and ideas are well structured and clearly presented, which greatly helps the reader to follow and understand the paper.
* The experimental setup is technically sound and the authors provide a thorough comparison and evaluation on different models and data splits (i.e. BLIND similar to prior work and PROBE, which targets specific probes).
* The selection of languages (i.e. English, Spanish, and Swahili) is also worth pointing out, as they represent distinct morphological features.

**Reasons To Reject:**

I did not identify a potential reason to reject the paper.

**Reproducibility:**

4: Could mostly reproduce the results, but there may be some variation because of sample variance or minor variations in their interpretation of the protocol or method.

**Reviewer Confidence:**

2: Willing to defend my evaluation, but it is fairly likely that I missed some details, didn't understand some central points, or can't be sure about the novelty of the work.

**Typos Grammar Style And Presentation Improvements:**

Figure 1: I find hard to distinguish between different shades of the same colour, and specifically, the sw-1PL and sw-FUT you mention in line 252.

---

> ### Author Rebuttal · Authors · 2023-08-26
>
> Thank you for your review.
>
> Regarding both reproducibility and clarity, we will publish our cleaned data sets as well as the splits that we used on Github if the paper is accepted.
>
> Regarding tagging inconsistencies, in Swahili, we find that the tags for the same inflection are sometimes presented in a different order, and sometimes they use different abbreviations for the same feature (e.g., PFV vs PRF for perfective). As part of the published data set, we will explain what cleanup we did in the README.
>
> The number of experts involved in our data preparation is the same as the number of authors on the paper. A small number which will be clear once the paper is deanonymized. Even twice that number would be "a large number" in our estimation, but still well down in the single digits. This is nice, because it means the effort to create larger and more varied probe data splits from existing UniMorph is very feasible.
>
> If accepted, we intend to spend most of the 5th page better explaining the probes and their purpose as well as the implications of the analysis. We have some vague intuitions about why ENC-DEC underperforms relative to the other models that we can add into the paper. The ENC-DEC model seems to struggle with keeping track of the contribution of the individual features in the feature sets, so it often makes strangely lateral guesses when just a single one is altered. Notably, the only Spanish probes that it succeeds on at all are the lexical ones involving conjugational class, not probes that held out features or sets of features, which further suggests that it's the feature sets that it struggles with. It was originally published in a paper that focused on English, which only has small feature sets, so this issue didn't really crop up.
>
> We will work on improving the color differentiation for Swahili in Figure 1. The colors were hand-selected, so they won't be hard to modify.

---

### Meta-Review · Area_Chair_btkL · 2023-09-19

**Recommendation:** 4

**Metareview:**

The short paper proposes a set of 13 language-specific (English, Spanish, Swahili) probes for morphological generalization that are based on conjugation classes.

All reviewers agree that the work presents a nice, well-written linguistically-motivated study and strongly suggest it for acceptance. I do not see any clear objections and/or potential risks and would be happy support the paper and see it accepted.

---

### Decision · Program_Chairs · 2023-10-07

**Decision:**

Accept-Main

**Comment:**

The short paper proposes a set of 13 language-specific (English, Spanish, Swahili) probes for morphological generalization that are based on conjugation classes.

All reviewers agree that the work presents a nice, well-written linguistically-motivated study and strongly suggest it for acceptance. I do not see any clear objections and/or potential risks and would be happy support the paper and see it accepted.